# Perspectives about social support among unmarried pregnant university students in South Africa

**Thandiwe Msipu Phiri, Patrick Nyamaruze, Olagoke Akintola**  *

School of Public Health, Faculty of Community and Health Sciences, University of the Western Cape, Cape Town, South Africa

* oakintola@uwc.ac.za

## Abstract

### Background

Pregnant young women in an academic environment are susceptible to stressors associated with unintended pregnancy and academic demands of universities. The challenges they face may be exacerbated by lack of social support, putting them at risk of psychological disorders such as depression. Women who receive social support during pregnancy and postpartum experience less emotional distress and report greater maternal satisfaction. However, very little is known about the perspectives about social support among unmarried pregnant students in tertiary institutions.

### Methods

Participants were purposively selected among unmarried pregnant students and those in the puerperal period at the time of the study. We conducted semi-structured qualitative interviews to explore the perspectives of unmarried pregnant students on the type of support that they need during pregnancy and the puerperium and the period when most support is needed. The data were audio-recorded and transcribed verbatim, then analysed using thematic analysis.

### Results

The findings show that social supports (emotional, instrumental, informational, and financial) were highlighted as important resources to cope with stressors during pregnancy and post-birth. Emotional support from male partners was the most important type of support needed as it entailed a sense of being loved and cared for. Social support was identified as important throughout the different phases of pregnancy and post birth, with different support needs expressed at each of these phases.

### Conclusion

This study identified support needs of unmarried pregnant university students in their transition to motherhood. Given the several challenges that they are faced with, unmarried

**Data Availability Statement:** All relevant data are within the paper and its Supporting Information files.

**Funding:** The authors received no specific funding for this work.

**Competing interests:** The authors have declared that no competing interests exist.

pregnant students need social support, including male partner support to enhance wellbeing as they try to cope with academic and pregnancy-related stressors.

## Background

Pregnancy marks the beginning of the transition to motherhood, and it is a period of great physical and psychological changes for all women [1]. For a lot of women, it is a period of time that is spent in material and psychological preparation for the arrival of a baby, whose presence brings about several changes in the mother's life. But for some women, this can be a period characterised by considerable physical and emotional stress. The stressors range from physical exhaustion, overwhelming tasks, decreased financial resources, social isolation and depressive symptoms [1, 2]. Research shows that young women experience several unique stressors during pregnancy due to their age and social context. Stressors such as academic demands, lack of income and relationship problems with their partners could affect their psychological wellbeing [3, 4]. Pregnant mothers therefore need support that involves undertaking tangible acts, showing concern, encouragement and care which increases potential for the expectant mother to embrace pregnancy-related changes and promote positive wellbeing for both mother and child [5, 6].

Although social support is of great value for all mothers during pregnancy, certain groups of women are especially vulnerable during pregnancy, and these include adolescents, unmarried women, students and women of low socio-economic status [7]. Because of the risky sexual behavior among students, young women in institutions of higher learning have a high risk of unintended pregnancy [8], particularly among unmarried students [8–10]. While some students may choose to terminate their unintended pregnancies as a legal right stipulated under the Choice in Termination of Pregnancy Act in South Africa [11], abortion is often publicly condemned and associated with negative and judgmental attitudes from health care providers and the community [12, 13]. Students that choose to keep their unintended pregnancies remain in need of a supportive environment that favours their physical and psychological wellbeing.

Young mothers have less support and face more challenges in terms of adjustment to motherhood as compared to older women [14]. For young mothers in an academic environment, the integration of their roles as students with their new role of motherhood can be confusing and stressful [15]. The challenges they face may be exacerbated by a lack of social support, putting them at risk of psychological disorders such as postpartum depression [16]. Furthermore, one study showed that married women received more support from their partners as compared to unmarried women [17]. This shows that marital status influences the amount of support that pregnant women receive from their partners. Women who receive support from their partners both during pregnancy and in the postpartum period have been shown to experience better mental health outcomes including less anxiety, emotional distress, depressive symptoms and report greater maternal satisfaction [18, 19].

University students are a population that is susceptible to stress given their academic demands [20]. Most university students are at an age and social context where they are transitioning from adolescence into young adulthood. This can be a difficult period particularly for first year students who struggle with fitting in, maintaining relationships and getting good grades amid the growing demands of life at the university [21].

Social support for unmarried pregnant women can come from different sources among them family, friends, male partners and their community. House [22] defines social support as

perception and actuality that one is cared for, has assistance available from other people, and that one is part of a supportive social network. Social support, either received from people closest to a person or offered through organizational interventions, has a positive influence on pregnancy experiences [23].

Social support is a multi-dimensional construct and has been conceptually divided into four domains: informational; instrumental; appraisal; and emotional support [24]. Emotional support includes any actions of caring towards and improvement of esteem in the recipient of the support. Emotional support for pregnant women could decrease the likelihood of psychological stress disorders and depressive symptoms. Informational support is generally advice and guidance given and, in this case, it could positively influence decisions on issues such as prenatal care, recommended nutritional and healthcare practices and preparation for labour and delivery [24]. Instrumental support involves giving help with tasks or providing material or tangible goods to an individual [24]. Instrumental support such as assistance with household chores and childcare could help expectant mothers cope with physical exhaustion and physically taxing demands which could cause strain, particularly in the last trimester of the pregnancy [25]. Appraisal support is providing evaluative feedback to others and consistent positive feedback during pregnancy and after birth may help normalise the concerns of mothers and build their self-esteem and self-confidence in their parenting role [26]. Support networks have a buffering effect on the stressors associated with pregnancy and motherhood [27]. In the context of maternal and child health, supportive relationships may have a positive effect on pregnant women, giving them a sense of personal control and enhanced feelings of wellbeing that would help them perceive their pregnancy as less stressful [27, 28].

Unmarried pregnant students are a vulnerable group considering the circumstances that surround their pregnancy such as their age, unstable relationships with their baby's fathers, unemployment/lack of income, academic commitments and social stigma. However, the support needs of unmarried pregnant students in university settings has received little attention. Therefore, the aim of this study is to explore the perspectives about social support among unmarried pregnant students in a university setting. The findings could provide insight into factors that could assist in designing interventions for this population.

## Objective

We sought to achieve two objectives in the study:

1. To explore the social support needs of unmarried students in tertiary institutions during pregnancy and the puerperium.

1. To explore the period when most support is needed by unmarried students in tertiary institutions during pregnancy and the puerperium.

## Methods

### Study design

The descriptive qualitative research design was seen as best suited to answer the research questions in this study. Descriptive qualitative research design enables researchers to get an insider's view of participants and their experiences in a natural setting [29]. The rationale for the use of a descriptive qualitative research design is to provide straightforward descriptions of experiences and perceptions [30], particularly in areas where little is known about the phenomenon under investigation. The choice of a descriptive qualitative research design was primarily determined by the type of questions the study sought to address, that is, to gain a deep

understanding of the type of social support needed by unmarried pregnant students. This design was appropriate as it fostered maximum co-operation and closeness between the interviewer and the participants and enabled them to describe their perspectives about pregnancy and support needs while studying in university.

## Participants

Participants in the study were all female students at a public university in South Africa. Participants were recruited if they met a set of inclusion criteria which included that they: were currently registered as undergraduate or post-graduate students at the university; were currently pregnant or in the puerperal period (less than six weeks post childbirth) at the time of the study; self-identified as being unmarried; and were willing to answer questions regarding their perspectives about their pregnancy and support needs.

Because issues relating to pregnancy among young people are sensitive and therefore not openly discussed, it was not easy to recruit participants for the study. Therefore, we used the snowball sampling technique in this study. This is a type of purposive sampling that allowed participants to recommend others in their circle who met the inclusion criteria and were willing to participate in the study [31].

The first author (TMP) initially approached three unmarried pregnant students who stayed in the same residence as her. She (TMP) discussed about the study with those students and requested them to identify and recommend other students who were unmarried and pregnant or had recently given birth. Next, she (TMP) approached and invited those who met the criteria to participate in the study. The first author was a Masters student and a young woman who had a pregnancy and delivered a baby while studying for her Masters degree. She had received training on qualitative interviewing as part of her Masters degree and additional training from the third author (OA) who is a Public Health Promotion and Research Methods specialist. She conducted all the interviews drawing on her own pregnancy experiences. The rationale for that was that some of the participants might be ashamed to express themselves when being interviewed by the opposite sex or an individual who had not experienced pregnancy. It has been shown that the gender of the interviewer can substantially affect the response rate in data collection [32].

Thirty-three students were invited to participate in the study and seven of them were excluded because they were past the puerperal period. Two of those invited to participate mentioned that they were not comfortable to discuss their pregnancy experiences and the remaining 24 were recruited to participate in the study.

## Data collection

Ethical approval for the study was obtained from the Humanities and Social Sciences Research Ethics Committee of the University where the study was conducted (HSS/0584/015M). In-depth interviews were conducted using an interview schedule which contained open-ended questions. The interview schedule was designed to suit the context of the study setting and to encourage unmarried pregnant students to discuss their experiences. The schedule was also reviewed and approved by the third author (OA) who was the research project supervisor. The questions contained in the interview schedule included the socio-demographic variables of participants and their partners, the support needs of unmarried pregnant students, the period when most support is needed and the role of their male partners. Some of the interview questions posed to participants include: "Describe some of the challenges you face in your pregnancy? As a pregnant student?"; "what stage of pregnancy do you feel you need the most support from significant others? Why and from whom?"; and "From the support needs you have already mentioned, which ones do you feel are the most important to you? Why?".

The interviews were conducted in a private and quiet seminar room within the university. This gave the participants an atmosphere of safety and comfort in which they could talk about their personal experiences considering the sensitive nature of the topic. The interviewer (TMP) made field notes during the interview and then summarized these following the interview. The field notes were thereafter used to develop analytical memos. Each interview lasted approximately 40 minutes. The open-ended questions provided the participants with an opportunity to interact freely and to discuss their perspectives regarding support needs during pregnancy in detail. Written informed consent was sought and obtained from all the participants before the commencement of interviews. Permission to record the interviews was sought and obtained from participants prior to the interviews. We used a digital audio voice recorder and the recordings were safely stored in a password protected computer. Data collection was carried out in accordance with relevant guidelines and the participants could withdraw from the study at any point. Although the study was of a sensitive nature, we did not anticipate any negative effects on the participants. However, we had arranged counselling services for the participants if the need arises but none of the participants requested for these services.**Data analysis**

The audio-recorded data was transcribed verbatim by the first author (TMP). The anonymity of the participants was maintained by assigning a pseudonym to each. We chose thematic analysis as the method of analysis for the data because it allowed us the opportunity to conduct iterations through careful reading and re-reading of the data in order to discover underlying meanings and patterns and to produce a detailed account of the phenomenon [33, 34]. As qualitative descriptive research is purely data-derived [35], we specifically employed an inductive approach to thematic analysis [36].

The first author (TMP) performed the initial round of coding following the six steps described by Braun and Clark [36]. The first step in the process was familiarization with the data by reading and re-reading the transcripts to make summaries. Secondly, she (TMP) generated initial codes and this step was followed by identifying the emerging themes, which is the third step. Although 16 interviews already indicated a point of saturation, we decided to continue analysis of the remaining interviews to ensure that no potential new codes were identified [26]. Fourthly, TMP reviewed potential themes, particularly checking for inconsistencies and whether the themes overlapped. Next, the themes were defined and named. Lastly, a report was produced using quotations of what the participants said to illustrate the themes [36].

It is important to note that this process was not straightforward as presented rather it was iterative. The initial coding was then reviewed and modified by the third author (OA) after which the second author (PN; a post-doctoral fellow and researcher), conducted a general review of the themes. All the authors revised, finalized and agreed upon the themes.

## Findings

In total, twenty-four students who met our criteria participated in the study (Table 1). Among these, twenty-one were pregnant and three were in the puerperal period. All participants, including those who were in the puerperal period had disclosed their pregnancy to significant others.

A summary of the characteristics of the participants are provided in Table 2. Four of the participants had one previous pregnancy while it was the first pregnancy for the remaining twenty participants. The findings are organized into four main themes and ten sub-themes identified from the analysis of the data. The four main themes are: Support needs of unmarried pregnant students; period when the most support is needed; male partner support required; and health systems role on social support.

**Table 1. Socio-demographic details of participants (n = 24).**

| Participant pseudonyms | Age | Gestational stage | Number of previous pregnancies |
|---|---|---|---|
| Mbali | 20 | Seven months pregnant | None |
| Lindiwe | 23 | Eight months pregnant | None |
| Busi | 19 | Four months pregnant | None |
| Samantha | 19 | Seven months pregnant | None |
| Nomthandazo | 19 | Four weeks puerperium | None |
| Bongiwe | 21 | Three months pregnant | None |
| Bella | 26 | Six months pregnant | One |
| Kholeka | 21 | Eight months pregnant | None |
| Zevile | 23 | Five months pregnant | None |
| Rachael | 22 | Five months pregnant | None |
| Thuli | 21 | Eight months pregnant | None |
| Njabulo | 24 | Six months pregnant | None |
| Khethiwe | 24 | Four months pregnant | One |
| Slindile | 20 | Three weeks puerperium | None |
| Amahle | 21 | Seven months pregnant | None |
| Khanyisile | 22 | Nine months pregnant | One |
| Thandeka | 20 | Four months pregnant | None |
| Lerato | 21 | Five weeks puerperium | None |
| Zinhle | 23 | Six months pregnant | None |
| Tamara | 20 | Three months pregnant | None |
| Naledi | 26 | Seven months pregnant | One |
| Siyanda | 19 | Four months pregnant | None |
| Nandi | 23 | Seven months pregnant | None |
| Ntando | 22 | Three months pregnant | None |

## Support needs of unmarried pregnant students

Support during pregnancy was perceived to be very important among all the participants. All the participants said they could not do without social support. The support needs of the participants were similar in many ways but sometimes unique to their situation. The following three sub-themes emerged from the interviews: emotional support, instrumental support, and informational support.

**Table 2. Summary of the socio-demographic details of participants.**

| Characteristics | | Number of participants (Percentage) |
|---|---|---|
| **Age of participants** | 15–19 | 4 (16.7%) |
| | 20–24 | 18 (75%) |
| | 25–30 | 2 (8.3) |
| **Year of study** | 1st year; undergraduate | 7 (29%) |
| | 2nd year; undergraduate | 9 (37.5%) |
| | 3rd year; undergraduate | 5 (21%) |
| | Honours; post-graduate | 3 (12.5%) |
| **Gestational age** | 1st month-birth | 21 (87.5%) |
| | Birth-6 weeks puerperium | 3 (12.5%) |
| **Number of previous pregnancies** | None | 20 (83%) |
| | 1 | 4 (17%) |

**Emotional support.**   Some of the participants indicated that they felt a sense of shame because of their pregnancy. Other participants mentioned experiencing feelings of shame and embarrassment at university and in their communities as a result of their pregnancy. This was exacerbated when people would stare at them and that was considered to be a negative attitude.

Some of my friends ignore me due to the pregnancy because they don't wanna walk with me to school and I also see myself as totally different from them. At first, I felt so neglected now I have accepted that am pregnant and things won't be the same. I usually walk alone. Some friends will support me, some will not. [Lindiwe, eight months pregnant, undergraduate]

Emotional support was considered to be the most important type of support among participants. Some participants expressed their need for a 'listening ear' and the desire to be understood and not be judged for getting pregnant. Their accounts showed that they experienced feelings of sadness and loneliness and considered themselves vulnerable because they got pregnant whilst pursuing their studies. Personal vulnerabilities and situational constraints led to loneliness among the participants. Due to loneliness, the unmarried students in this study expressed a need for social relationships that would offer constant assurance that someone was there for them. Therefore, to minimize these feelings of loneliness, emotional support was considered to be the most important type of support that they needed to receive among all others.

It feels good to have people around and knowing that you are loved. Although you made mistakes—not that a baby is a mistake–but being pregnant while you are studying, being something that you have not planned obviously. So, knowing that although you have made that mistake, but people haven't abandoned you. People around you like your loved ones and your family- that they still love you. [Slindile, three weeks pueperal, undergraduate]

In comparison to financial support, receiving emotional support and having people who were going to be there for them in times of sadness was said to be important and it made their pregnancy easier to handle.

I just don't need money in that kind of way. The kind of support that I would really like, that I would really expect is for people to be there for me, to give me that much attention, to be there when I feel sad. [Rachael, five months pregnant, undergraduate]

For some participants, having someone available to offer emotional support was important regardless of who was offering it. They reported that the availability of emotional support made a huge difference in their lives regardless of its source because their greatest need was to feel loved and cared for. Having people to talk to and understand them, without judging them was important for participants.

A person who is pregnant just needs support, love—no matter how bad the situation is or how it has happened. Having people to talk to and that understand, are really there for you, lending their ear more than their mouth, people to talk to. [Samantha, seven months pregnant, undergraduate]

**Instrumental support.**   Receiving help with tasks and help to manage their time was an important support need of some participants. Their narratives revealed that instrumental support was most needed at the end of the pregnancy because as the pregnancy progressed, it became more difficult to be mobile. Therefore, they needed people to be around and help them with chores, academic tasks, accompanying them to certain places especially hospitals or going to buy them food and groceries. One participant noted that she needed company every time she was going to the health facilities unlike before she got pregnant.

If I was sick and I wasn't pregnant I'd be like, "no please, don't come, I'll go by myself". Now! [that she is pregnant], Now I can't go to the clinic by myself, I'll just call anyone, call my sister or my aunt, she's always there she's like no I'll take you. If it [support] had to stop now, I don't know what I would do. [Amahle, seven months pregnant, undergraduate]

Another participant revealed how overwhelming the tasks had become in her last trimester of pregnancy and the importance of receiving instrumental support during this period.

When you are nearing your end [approaching delivery], you have so much to do. You have got school things to do, you have got doctors' appointments consistently one after the other, you have to buy this, get this, get that, that's when you need a more rigid [solid] support system. [Khanyisile, nine months pregnant, second pregnancy, undergraduate]

These excerpts show that apart from the need to feel better about their situation, unmarried pregnant students need help with tasks to relieve them of physical stress. In addition, receiving help with tasks also had an effect on their emotional wellbeing.

**Informational support.**    The narratives of the participants in this study revealed that young women, particularly first-time mothers are usually inexperienced with pregnancy and childbirth and need advice and information. Some participants reported that they needed people around them that could help them with information about pregnancy, childbirth and how to care for their baby.

I do not have experience on how to bath the baby, feed the baby well when it's crying, what's wrong. They [people] should tell me what to expect if am not figuring out, they must assist me. [Busi, four months pregnant, undergraduate]

Other participants also indicated that they needed information on what to expect during the different stages of pregnancy and childbirth. One of the participants who had a previous pregnancy explained the difficulties of having a first pregnancy without adequate knowledge of the accompanying physical changes.

At the beginning of pregnancy, maybe you are not sure what is happening to you, may be for some people it's the first time you don't know what is happening, all the sickness. At the end that's exactly what's happening to you, breaking water and all that. Some of the things you wouldn't know so someone should tell you. [Bella, six months pregnant, second pregnancy, post-graduate]

Participants require guidance on pregnancy with respect to proper nutritional practices, adequate prenatal care and preparation for labour and delivery. In a unique case, one of the participants was at risk of adverse pregnancy outcomes as she lacked adequate information on proper nutrition practices as illustrated below:

After I fell pregnant, I did not eat. I don't usually eat. I didn't know that when you fall pregnant you have to change your lifestyle and do things in a certain way. It was a shock when I went for a scan when I was five months pregnant and they said my baby looks like he's two or three months, he was underdeveloped. So, the doctor said "it's either your child is gonna be disabled or when you give birth, you're gonna have to stay in hospital for a long time cause he's really small". [Samantha, seven months pregnant, undergraduate]

The participant's narrative highlights the challenges of poor nutritional intake during pregnancy.

## Period when the most support is required

When asked about the specific period during pregnancy when they feel the most support is required, participants had different views and four sub-themes arose.

**At the beginning of pregnancy.**    Some of the participants said they needed the most support, particularly emotional support at the beginning of the pregnancy. This was a period

when they had just found out about their pregnancy which was something new to them. At the same time, they were still dealing with acceptance of the pregnancy whether personally or in their relationship. Therefore, they indicated that at this initial stage of pregnancy, they required some form of assurance from someone that they were going to be supported.

I think I needed more support at the beginning. It was something new to me. I kind of needed someone to hold my hand and say "it's gonna be fine, I'm gonna be therefore you, don't worry, we'll get through this together". [Thandeka, four months pregnant, undergraduate]

In addition, this was the period when they were likely to be hiding the pregnancy from their family when in actual fact, they needed their support. Therefore, they found it difficult to share their stressors with others.

It was difficult before I told my parents because you know when nobody knows at home and you start experiencing these things [disagreements/fights] with your boyfriend and you get pissed, like you wanna tell someone but you can't. [Tamara, three months pregnant, undergraduate]

Therefore, it is important to receive support at the beginning of pregnancy for better maternal and new-born health outcomes. The different experiences that the participants had in their pregnancies determined what they perceived their support needs to be and the period that they were required.

**Throughout pregnancy.** Other participants said they required support throughout their pregnancy and into the puerperium because they felt their support needs were not going to change over the course of the pregnancy. Therefore, they said that they needed to receive support through every period of the pregnancy.

Throughout the pregnancy you need support because there are times where you feel like I wish I could just give up everything like you know, school cause sometimes you find it frustrating, and you don't wanna deal with it. I think you need that pillar of strength every day; someone that will be calling you and asking you 'How are you doing? How's everything? How's the baby?' Like I think every day during pregnancy even after pregnancy. [Njabulo, six months pregnant, post-graduate]

**Towards the end of pregnancy.** Some participants felt they needed more support as they approached their delivery date. The support required at this stage was mostly instrumental support. They indicated that it was difficult to balance antenatal care (ANC) schedules at the public clinics and academic obligations in the final trimester. This suggests that in the last trimester of pregnancy, students experience a task overload. For this reason, they need instrumental support in the final trimester of pregnancy.

It was at the end [when she needed more support]. When you feel like the baby is about to come, you have to prepare for everything. With schoolwork it's very hard. Because at the end, I had to attend clinics more than before, like almost every week. So, the time you are attending clinics and everything, you are using up the time for studying. [Slindile, three weeks puerperium, undergraduate]

One participant expressed her need to always have someone with her so that she would be well prepared in the event of labour.

Mostly it's at the end when you'll be delivering (when she needs the most support) because at that time you have a lot of complications, you need someone to look after you because you can deliver anytime. You need to have transport that will take you to the hospital immediately and always bring your card so that you are well prepared. [Thuli, eight months pregnant, undergraduate]

**Support post-delivery.** Young women usually require support and guidance after childbirth and in the case of students; they need support with childcare as they continue with their studies. Except for two who were not sure, other participants expressed their intention to continue with their studies after their child is born. When asked about childcare, some participants' responses were that their mothers or female guardians were going to look after the baby upon delivery. In cases where mothers were unavailable due to distance or employment, they were still going to offer support by paying a helper to look after the child. Therefore, parents were also going to provide financial support to a large extent. Young women in this study relied heavily on their own mothers and depended on them for childcare.

I'll still be in [city name] when I give birth cause am giving birth in September. So, my mom will travel here. My mom said "give me the child then you go back to school" and she's gonna take care of the baby...In terms of finances my mom will support me. I don't know about my child's father whether he'll be there or not. [Thandeka, four months pregnant, undergraduate]

Those who did not have mothers (due to death) or those whose mothers worked faraway would receive help from other older female members of the family such as aunties and sisters.

My mother's sister, she lives in Maritzburg [Pietermaritzburg], she did say she can take care of baby once the baby is born so that I can go back to school and study cause that is what I want to do. She was the one who volunteered to do that. [Busi, four months pregnant, undergraduate]

When participants were asked about financial support, some of their accounts revealed that providing for the child's welfare was the father's responsibility.

The baby needs so many things so as to grow in a good state; nappies, warm clothes, milk, formula [baby food] since am not going to breastfeed so financially it will be much required from him. [She gives an example]-cause my cousin gave birth in April and the father of the baby hasn't given her a single cent since she gave birth, since she fell pregnant, she hasn't received any money from the baby's daddy and telling me things are difficult, so financially it is so important. [Kholeka, eight months pregnant, undergraduate]

## Male partner support needed

One universal theme that emerged in all the interviews with respect to support needs was receiving support from the male partner. Male partner support was considered to be the most important source of support that unmarried pregnant students required. The lack of emotional support from male partners during pregnancy was seen to have grave consequences for both the mother and her unborn baby. These include depression, resentment and abortion.

As much as you are the one carrying the child, you can't go through everything by yourself. I think if you don't have your partners' support, you can end up having resentment towards him, including the child—somebody who has no contribution towards the baby is painful. [Khanyisile, nine months pregnant, second pregnancy, undergraduate]

**Partner's emotional support.** Receiving emotional support from partners was the biggest support need among young women. When compared to other support needs, some participants reported that they would rather have their partner to be there for them emotionally than financially. The emotional support required by these participants ranged from the simplest acts of care like being there to talk to them or calling them every day.

I would like to be supported emotionally because money doesn't buy love. It is the tender loving care that is most important. I appreciate that because that is what is important, that is what I prefer, it's what I need. [Bongiwe, three months pregnant, undergraduate]

Other participants said that their partners needed to be their greatest source of support because they were the reason why they got pregnant. Therefore, they needed to take responsibility for 'what they did' and 'compensate for their mistake'. This means that male partners were expected to play a major role in providing support and relieving emotional distress.

You know when you are pregnant, I think you expect a lot when you know that this person is the baby's father, it's like you want them to do more. You want them to try anyway and compensate for doing what they did to you, because if it wasn't for him I actually wouldn't be pregnant so he has to be there for me. [Zinhle, six months pregnant, undergraduate]

There was a general view among some participants that the relationship with their partner should remain as it was before pregnancy or improve with the prospect of a child. However, it was found that in a lot of cases, relationships became weak as the pregnancy progressed. In some cases, the pregnancy led to the end of the relationship. Others revealed that relationships become more about the wellbeing of the unborn baby and less about the romantic relationship between the partners. Some participants reported feeling neglected and expressed longing for a romantic relationship as they did before the pregnancy.

I don't want him to treat me as the mother of his babies, I want him to treat me also as his girlfriend. Now we usually talk about the baby a lot. We don't talk about us. I want us to date like we used to before, go out with him, watch some movies and get a lot of support from him. [Thuli, eight months pregnant, undergraduate]

**Partner's instrumental support.** Having a partner who is available to provide material goods and help with tasks was important among the participants. Instrumental support such as having the partner's company when they go places and buying them foods that they crave for was considered to be an important support need. The smallest acts of help with tasks were considered to be of great importance even in cases where partners could not offer financial support.

Sometimes people ask questions about him [her partner] when you are walking alone. I need the support. If he was not there for me, it would have hurt me you know, to know that people are laughing at me. [Kholeka, eight months pregnant, undergraduate]

One participant complained about her partner who is not usually willing to help her with tasks.

Sometimes he'll be sleeping [when he visits her in her campus room] and I have to do my schoolwork then when I ask him to do something like can you please get me some water, sometimes he's gonna be like he's tired or when I want him to do something for me he's gonna complain. [Zinhle, six months pregnant, undergraduate]

Instrumental support was considered to be crucial particularly in matters involving doctors' appointments and preparations for the arrival of the baby. In cases of medical emergencies, partners were required to be present. Going to the hospital together was considered to be an important aspect of instrumental support as it assured them that they were not going through their pregnancy alone.

Before I fell pregnant, I really didn't notice anything, and it didn't matter. Now it's more like I want more, more time with him. Sometimes when you go to the clinic and you see people with their boyfriends and then I'll be thinking like, where's mine? I also want him to be here and experience this together. [Amahle, seven months pregnant, undergraduate]

**Partner's financial support.** All the participants said that their tuition is paid by their parents. Therefore, the financial resources expected from male partners were for issues concerning the pregnancy. Financial needs from partners included money to buy food that they

crave, maternity clothes, hospital bills and items for the baby. However, more financial support was expected after delivery as compared to during pregnancy.

I need him not to run away from his responsibilities when the baby is here, that's all I'm just hoping for, that he'll be a father and support his child. [Lindiwe, eight months pregnant, undergraduate]

Financial support did not always mean receiving money from their partners. Having their partners buy or pay for their needs was also regarded as financial support.

Well, my partner is not good at giving money and all that, but he can do everything and anything to support me, he put me on his medical aid [medical insurance], he can buy me anything that I crave for, but giving money no. So, he supports me, yes. [Naledi, seven months pregnant, second pregnancy, post-graduate]

The payment of damages (Inhlawulo- in Zulu law, is a fine paid to the family of a woman who became pregnant out of wedlock by the father of the future child) was considered to be a form of financial support as it was money that could be used by the family to support the pregnant women. This study found that some of the participant's partners had not paid this fine to the young woman's family. In the current study, men who paid damages to their partner's family were perceived to be more responsible and could be entrusted with caring for them. The payment of damages was observed to have a calming effect on angry parents and associated with a more positive attitude towards the pregnancy. Therefore, it was considered as a type of support that was required of partners to assure the family that he would honour his responsibility.

My sister is the one who advised me about [partner] paying the damage in December because that is when my dad is coming from Joburg. I want him (father) to know that even if he chases me [from his house], he will know that everything was done perfect and I'm not staying with, what can I say, a thug or anything. I want him to know that am staying with the real man. [Mbali, seven months pregnant, undergraduate]

On the other hand, men who did not pay damages were seen as untrustworthy.

To be honest sometimes I think he is going to run away, since he hasn't paid damages and he can see the other girls, and forget about me you know, you know how boys are. [Nomthandazo, four weeks puerperium, undergraduate]

## Health systems role on social support and influence on male partner support

Health professionals are able to contribute to the psychological wellbeing of women during pregnancy and during the postpartum period by offering support. When asked about their experiences at ANC clinics, some participants reported that the health staff at the clinics had a negative attitude towards them. They felt stigmatized because of their age and status as students, and this was because of what the nurses would say about them.

Ooh at first, I had hell, like the nurses ask you basic questions: "how old are you?" Then when you say 18 then everybody is like "hau?" then instantly when you say you go to [institution name] everybody assumes like "no wonder, it's expected". [continues to mimic] "Where's the baby's daddy?" Am like "he's working". They're like "yeah, you have a working baby's daddy because that's what you do". They weren't really saying it to my face, but you can hear them gossiping. [Siyanda, four months, undergraduate]

When asked about how the health systems influence male partner support among pregnant students, particularly partner ANC attendance, almost all the participants said they had never attended an ANC session with their partner. Some participants said their partners did not want to go to the clinics with them because of the female dominated environment at the clinic and the amount of time spent at the clinic. In addition, nurses did not seem to be concerned about the absence of partners.

I wanted him to be there, but he suggested that no, he will remain behind, or he will go with me and leave me there. Personally, I understand why he doesn't want to stay with me. Clinics are just something else, many people and you wait for hours. The nurses don't even ask about my partner. [Kholeka, eight months pregnant, undergraduate]

Some participants explained that they were not attending ANC in public clinics but instead were consulting specialists like gynaecologists and obstetricians in private hospitals to monitor their pregnancy. This was mostly because they felt that they needed to be checked by specialists who were not readily available in public clinics.

With my partner first of all he said he doesn't want me to go to clinics and all that, so he put me on his medical aid [medical insurance]. He didn't want me to be going to the clinics. There's nothing wrong with the clinic but for me he preferred the gynaecologist who is gonna do everything. [Naledi, seven months pregnant, second pregnancy, post-graduate]

Some participants mentioned that their partner did accompany them to the clinic on first visit but did not do so on subsequent visits. While they had the desire to have their partners come to the clinics with them, they mentioned that they did not see the purpose because nurses at the clinic did not involve men in any of the activities.

He came with me to the clinic on my second appointment, when I was about four months. It means a lot to have someone with you. But they didn't ask him any questions or asked him to do any tests. [Slindile, three weeks puerperium, undergraduate]

Overall, it was established that most unmarried pregnant students seek ANC in public clinics and hospitals alone without the support of their male partners. The participants generally considered ANC to be unfriendly. In addition, partner attendance was not specifically encouraged or recommended by healthcare workers.

## Discussion

Unmarried pregnant students need emotional, instrumental and informational support. Research on social support has consistently found that the absence of social support is a significant risk factor for antenatal and postnatal depression and anxiety among pregnant women [37, 38]. This study sought to explore the support needs of young unmarried students in an academic setting during pregnancy and the puerperium. Our findings suggest that the sociocultural context of pregnancy among young unmarried students results in social stigma. Unmarried pregnant students are stigmatized in the academic environment as well as in the community. This study found that these students are sometimes ignored by their peers because of the pregnancy. This leads to social isolation, a common effect of pregnancy among young people [39]. Social isolation has been associated with loneliness, emotional distress and depressive symptoms. Experiencing depressive symptoms in pregnancy has consequences for maternal and child health outcomes because it is associated with low birth weight [40].

Four different types of social support (emotional, instrumental, informational, and financial) were highlighted as an important resource to cope with stressors by all participants. Although financial support is usually categorised as instrumental support, in this study we

categorised it as an independent form of support as it was a salient theme in the study. Emotional support was described as having people who are available to share their feelings with, encourage them and show acts of caring towards them. Instrumental support expressed by participants included receiving help with tasks such as going for medical check-ups. The participants in our study indicated that they require informational support, which is receiving information on pregnancy, childbirth and how to take care of a new-born baby. Lastly, financial support was needed to buy food, to cover transport costs and medical expenses and to prepare for the arrival of the unborn baby. All these types of support play a role in fostering positive maternal and child health outcomes. However, appraisal support did not emerge as an important source of social support in this study and this could be because most of the participants were still pregnant and felt they did not need evaluative feedback. Yet, this form of support could have been of great value to them after delivery as they engage in motherhood duties such as feeding and bathing the infant.

An important finding of our study is that, among the different types of social support needed, emotional support was deemed more important by the participants. Pregnancy was associated with loneliness and fear of being judged for falling pregnant within an academic setting. The young women desired to be listened to and loved more especially because of their condition which was associated with various stressors. Although a previous study reported on the importance of emotional support during pregnancy, it was not regarded as the most important type of support needed [41]. The provision of re-assurance, feeling valued and sympathetic listening result in feelings of comfort and is associated with decreased levels of depressed mood and anxiety during pregnancy and postnatal [26]. In line with Cohen and Will's [42] stress buffering hypothesis of social support, unmarried pregnant students perceive emotional support to buffer the effects of psychological distress during pregnancy. Previous studies have also shown that perceived social support may have a positive effect on pregnant women causing them to perceive their pregnancy as less stressful and providing a resource for effectively dealing with stressful situations such as labor [43, 44].

For unmarried pregnant students, who usually stay at university residences, instrumental support was perceived as important in assisting to balance the academic demands and pregnancy-related demands. The pregnancy came with additional responsibilities such as regular visits to the clinic and performing previously manageable tasks was perceived as daunting as a result of the pregnancy. A study in Ghana among women with new-borns and various community stakeholders found that women were dependent on network members for instrumental support to get them to a health facility for childbirth [45]. However, instrumental support seemed unavailable to pregnant women residing at university residences as they did not stay with their families. The narratives of the participants revealed that young women, particularly first-time mothers were usually inexperienced with pregnancy and childbirth and need advice and information. In a previous study, 70% of first-time pregnant women indicated that they needed information about pregnancy and childbirth [46]. The participants expressed a lack of knowledge regarding pregnancy-related sickness, delivery process, and dietary needs as well as taking care of the newly born child. A recent study among adolescent pregnant women has shown knowledge deficit on the benefits of ANC and nutrition during pregnancy, that may contribute to poor pregnancy outcomes [47].

Our study sought to understand the stage at which social support was deemed more important by the unmarried pregnant students. Social support was identified as important at different phases of pregnancy. In the first few months of pregnancy, the expectant mother develops new and often uncomfortable physical and emotional symptoms such as nausea, feelings of uncertainty and increased emotional expressions [48, 49]. In the early stages of the pregnancy, young women desired support from significant others as they were still in the process of

embracing the pregnancy and thus required acceptance and assurance that they would be supported throughout the whole pregnancy experience. Towards the end of the pregnancy, informational support was deemed more important to help balance between academic demands and regular visits to the clinic. More so, the narratives from most of the participants indicate that support was necessary throughout the whole period of pregnancy and after giving birth. The various stressors faced by unmarried pregnant students came at different stages of the pregnancy and were accentuated by the challenging academic environments characterised by high workload, meeting deadlines and socio-economic issues [50, 51]. Thus, an enduring and reliable support network was essential.

While support from different sources is important for unmarried pregnant students, participants identified male partner support as their most important source of support. Unlike community-dwelling women in Bangladesh who cited mothers and sisters-in-law as top sources of emotional support [41], participants desired their partners to be emotionally available and care for them since they were partly involved in conceiving the child. Male partners are expected to take responsibility and be available to offer emotional support to young women throughout pregnancy and after childbirth. The availability of support from male partners gives women a more positive attitude towards their pregnancy and assures them that they are not alone thereby reducing emotional distress [19]. Partner support is an important protective factor throughout conception and pregnancy and our study corroborate these findings. Importantly, participants did not want to be perceived by their partners as mothers only, but as lovers, as that entailed emotional connection. This type of support and relationship status was deemed more important than financial support. A study among couples indicated that more prenatal shared leisure predicted higher marital love and less conflict when the baby was 1 year old [52]. Even though participants in the current study were unmarried it seems that they would benefit from spending more quality time with their partners. A previous study showed that male partners who did not approve an unplanned pregnancy were less likely to be emotionally available increasing the risk of psychological distress in women [3]. In instances where emotional support from male partners is not feasible, there is a need to design interventions that will assist unmarried pregnant students in identifying alternate sources of support such as counselling support, seeking social support from friends and their social circles as well as through the social media.

Partners' instrumental and financial supports were also considered to be important sources of support. The participants wanted to be accompanied to clinic visits and experience the pregnancy process with their partners. In previous studies, male involvement in maternal and newborn health has been associated with a variety of improved health outcomes including increased maternal access to antenatal and postnatal services, improved maternal mental health and reduced postpartum depression [53, 54]. In some African cultures, payment of damages is perceived as a form of acceptance of paternity [55]. This process was perceived by the participants to give a form of dignity to her family and some assurance that the father was prepared to take care of the child. However, similar to a previous study, expectations of partner support (emotional, instrumental and financial) are not usually met resulting in disappointment, conflicts with their partners and sometimes difficult break ups [56]. The current study found that experiencing an unintended pregnancy can lead to the disruption of a relationship. These findings complement the results of research showing that unplanned pregnancies are accompanied by family disruption and a decline in the marital quality and the couple's relationships [3, 57]. This results in emotional distress for unmarried pregnant students who may have to experience pregnancy without the support of their partner.

Unmarried pregnant students experienced social stigma at health facilities. This finding is consistent with the results from a previous study among healthcare providers in Ugu district

in southern KwaZulu-Natal, South Africa which showed that health personnel exhibit a negative attitude towards pregnant young women [58]. The South Africa Department of Health guidelines recommend that pregnant women seek ANC as soon as they suspect a pregnancy and encourages all pregnant women to attend at least eight ANC visits [59]. However, the experience of social stigma at public health facilities is likely to result in unmarried pregnant students shunning ANC, yet most of them cannot afford to use private healthcare services. Young women in our study also expressed the desire to have male partners attend ANC visits with them. However, it was reported that this does not usually happen due to the amount of time spent at ANC clinics, the female dominated environment and the negative attitudes from the nurses. These findings add to previous research which demonstrated that the main barrier to male participation in ANC was the healthcare facility environment [60]. Further, during ANC visits, nurses only ask about the existence of male partners but do not necessarily encourage partner attendance of ANC. Therefore, there is minimal influence from the healthcare professionals on emphasizing partner attendance of ANC, their perceived roles and benefit to the mother and the baby. Health systems have a role to play in fostering supportive relationships between pregnant women and their partners. Partner support and involvement in maternal healthcare has been shown to enhance maternal health outcomes [61].

Among women of all ages, pregnancy is associated with a myriad of stressors. In the contexts of academic institutions, pregnancy-related stressors are likely to add to the academic challenges faced by female students [62]. Absence or lack of emotional social support from significant others, particularly partners may exacerbate the stressors of pregnancy among students. Tertiary institutions should partner with established social initiatives offering services for men in the country to engage and mentor young men one-on-one and in groups about the importance of the father's role in the child's development and how to best support their partners to improve pregnancy outcomes, irrespective of relationship status. Culturally sensitive health promotion programmes targeted at 'father inclusive' practices and behaviour change communication programmes to stimulate social support may have positive changes in terms of the supportive role of partners [63]. To reach many young men, implementers of such programmes should consider adopting innovative outreach strategies that appeal to young men's interests in their hang-out spots, such as bars and sports arenas.

Male involvement programmes can be integrated in ANC whereby services are restructured to accommodate male partners to ensure a sense of involvement and responsibility. However, for these programmes to work, the ANC environment should be user friendly for young women and their partners. A rights-based approach must be consistently adopted when delivering maternal healthcare [64]. Training and on-going education should be prioritised for healthcare workers regarding the provision of positive, respectful and supportive non-judgmental antenatal and postnatal care to their clients irrespective of their age or marital status. Pregnancy at universities is not uncommon and having clinics at university campuses equipped with comprehensive ANC may overcome some structural barriers such as the burden of transport costs and long waiting times at public clinics.

## Strengths and limitations

To our knowledge, this is the first study to present a rich description of the perspectives about social support needs and the period when most support is needed by unmarried pregnant university students in South Africa. We also sampled until data saturation to ensure that a sufficient quantity and quality data was collected. However, the snowball method of sampling that was used meant that the initial participants, who were indigenous African students, recommended other pregnant students in their social network, who were also indigenous African

students. As a result, potential participants from other racial groups did not participate in this study. Therefore, the findings could be biased and be a representation of experiences and perceptions of pregnant indigenous African students. Also, the findings of this study apply only to pregnant students who are unmarried and does not apply to those pregnant students who are in a marriage or engaged to be married. Future studies should include different racial groups to capture the experiences of pregnant students of all races and also include married and unmarried students to compare and contrast their support needs.

## Conclusion

This study identified support needs of pregnant women in their transition to motherhood. Among the support needs of unmarried pregnant students, emotional support is perceived to be the most important type of support. Male partners appear to be the most sought-after source of emotional support for unmarried pregnant students as they desire to experience the emotional connection that existed before the pregnancy and fear the possibility of abandonment. Given the several challenges that they are faced with, unmarried pregnant students need all the available support to enhance wellbeing as they try to cope with academic and pregnancy-related stressors.

## Supporting information

**S1 File. Minimal underlying data.**
(PDF)

## Acknowledgments

The authors would like to express their sincere thanks to the participants for the value they added to the study.

## Author Contributions

**Conceptualization:** Thandiwe Msipu Phiri, Olagoke Akintola.

**Data curation:** Thandiwe Msipu Phiri.

**Formal analysis:** Thandiwe Msipu Phiri, Patrick Nyamaruze, Olagoke Akintola.

**Methodology:** Thandiwe Msipu Phiri, Patrick Nyamaruze, Olagoke Akintola.

**Supervision:** Olagoke Akintola.

**Writing – original draft:** Thandiwe Msipu Phiri, Patrick Nyamaruze, Olagoke Akintola.

**Writing – review & editing:** Thandiwe Msipu Phiri, Patrick Nyamaruze, Olagoke Akintola.

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
