## [Decision Letter · Decision Letter 0]

16 Nov 2022

PONE-D-22-16990

Experiences of social support among unmarried pregnant university students in South Africa.

PLOS ONE

Dear Dr. Akintola,

Thank you for submitting your manuscript to PLOS ONE. After careful consideration, we feel that it has merit but does not fully meet PLOS ONE’s publication criteria as it currently stands. Therefore, we invite you to submit a revised version of the manuscript that addresses the points raised during the review process.

I thank the authors for this very interesting and important manuscript. The reviewers have raised some important and extensive points, which I would like to see fully addressed before further consideration of this manuscript. Additionally, please read through the manuscript very closely for typographical and wording errors. 

We look forward to receiving your revised manuscript.

Kind regards,

Tara Tancred, PhD

Academic Editor

PLOS ONE

https://journals.plos.org/plosone/s/fileid=ba62/PLOSOne_formatting_sample_title_authors_affiliations.pdf.

4. PLOS requires an ORCID iD for the corresponding author in Editorial Manager on papers submitted after December 6th, 2016. Please ensure that you have an ORCID iD and that it is validated in Editorial Manager. To do this, go to ‘Update my Information’ (in the upper left-hand corner of the main menu), and click on the Fetch/Validate link next to the ORCID field. This will take you to the ORCID site and allow you to create a new iD or authenticate a pre-existing iD in Editorial Manager. Please see the following video for instructions on linking an ORCID iD to your Editorial Manager account: https://www.youtube.com/watch?v=_xcclfuvtxQ.

Additional Editor Comments:

Reviewers' comments:

Reviewer's Responses to Questions

**Comments to the Author**

1. Is the manuscript technically sound, and do the data support the conclusions?

Reviewer #1: Yes

Reviewer #2: Yes

2. Has the statistical analysis been performed appropriately and rigorously? 

Reviewer #1: N/A

Reviewer #2: Yes

3. Have the authors made all data underlying the findings in their manuscript fully available?

Reviewer #1: No

Reviewer #2: Yes

4. Is the manuscript presented in an intelligible fashion and written in standard English?

Reviewer #1: Yes

Reviewer #2: No

5. Review Comments to the Author

Reviewer #1: Research ethics: It is indeed important to observe research ethics but this does not justify your statement that 'data cannot be shared publicly because of privacy and confidentiality issues'. You can anonymize the transcripts and make them publicly available. Already you have used pseudonyms to protect the identity of participants (Line 210 and Table 1). You have correctly indicated (Line 166) that 'issues relating to pregnancy among young people are sensitive' but you have revealed the identity of the institution where data was collected (Lines 160, 161 & 163). I suggest that the name of the university be concealed.

Participants: The study is among unmarried pregnant university students as you clearly indicated in the title (Line 1) and some other places in the manuscript. I find it confusing when you refer to participants as 'pregnant women'; 'pregnant young women'; 'unmarried pregnant students'; young unmarried women' in Lines 46, 147, 321, 555 and at some other places. I suggest that a concept to refer to participants be identified and be used consistently. Looking at the age of the participants (Table 2), is it appropriate to refer to all of them as young women? I suggest that Table 1 be moved to Results as it reflects what the study found.

It is a good practice to describe the qualifications and experiences of researchers in a qualitative research. The description (Lines 174 - 179) of the first author (TMP) is correct, and I suggest that the other authors be described as well.

Language: Replace Master's (Lines 175 - 177) with Masters. School (Lines 242, 482) should be replaced with university. The manner of writing ANC (Lines 527, 536 and 551) should rectified. Write in full with abbreviation in brackets on first time use, then abbreviation thereafter. The concept 'clinics' (Lines 536 - 538) needs to be clarified as specialists run clinics in both private and government/public health services. Lines 551 - 557 regarding ANC need to be rephrased as ANC is available through private and public health facilities. Furthermore, pregnant women can deliver in a public hospital even when they did not attend ANC at a public/government facility or even when they did not attend ANC at all. Therefore, Lines 668 - 669 should be revised. Line 553 - 554 contradicts itself regarding affordability of private healthcare (none & except), see also Lines 670 - 671 regarding ability to pay for private healthcare. Pregnancy and conception (Line 639) should be conception and pregnancy.

Description of study design (Lines 150 - 158) should be rephrased to clarify if qualitative research is a design or a method. Based on what you wrote in Line 157, the appropriate design appears to be a narrative design or even a phenomenological design might be suitable.

Indicate under Data collection if field notes were collected or not as they are important to add to the context of the interviews.

Data analysis: Indicate (Line 209) that data was transcribed verbatim. This is only indicated in the Abstract (Line 39), but it is important to indicate this in the main document.

Results: Indicate the number of sub-themes in Line 229. Line 239 should say sub-themes, not themes. Clarify also 'four main subthemes arose (Line 334). Is the word 'main' necessary? The following concepts [many of the participants; several participants; most participants; a few of the participants] are not appropriate to report qualitative findings as they suggest numbers; appropriate concepts are some participates, other participants and others. Writing just 'participants' may also imply that it is all participants who shared that experience. I found both appropriate and inappropriate concepts in this manuscript and suggest that the inappropriate concepts be replaced. Line 311 should acknowledge that some participants had previous pregnancies and therefore knew what to expect. Lines 319 - 321 and 330 - 331 appears to be misplaced; they would suit the Discussion section.

Context: It is important to mention if these pregnancies were planned or not, if participants disclosed their pregnancies to whoever they expected to give social support and whether they are still in a continuing and healthy relationships with the partner who impregnated them. Where (on-campus or off-campus) and with whom were participants staying? Lines 599, 606-607 are not clear regarding where participants were staying.

Discussion: Lines 590 and 648-649 give an impression that types of social support were rated in this study. Rating is a quantitative concept. Lines 653-656 should say some African cultures because in some communities in South Africa, paying damages means admission that one has impregnated a women by paying a fine and not a commitment to support the child financially (paying damages is a once-off payment). Is Line 655 referring to all participants?

Recommendation: It is not clear on which findings are the recommendations on behaviour change programme (Line 688) and rights-based approach (Line 639) based. Who are the 'antenatal health personnel' in Line 695?

Reviewer #2: There is need to rephrase the topic to experiences of unmarried pregnant students on male partner social support to speak to the background of the study.

On abstract the section of methods line 35, the candidates say perceptions there must be consistency in usage of words, what is explored are experiences.

40-41 the results should be reported in line with the study aim which is pregnant women support needs and when it was needed the most.

Under line 150, it should be clear which research design under qualitative research approach was employed that informed the thematic content analysis.

Under 171 how many students were snowballed were in puerperal period.

Line 145 should specify the experiences from unmarried pregnant students are about "male social support".

119-144 provides an impression that the social support explored is only from male partners. Line 147 emphasis the said point which contradicts the study topic.

On line 197 the interview indicates that the schedule elicited the support needs of pregnant students and the period when most support is needed. The results should be aligned to what the interview guide elicited

on line 149, the section on study objectives should be included.

line 171-173, how many peripheral students were identified, only pregnant student are mentioned.

Under 189, what ethical codes were considered in this study and the study is sensitive in nature, how was the emotional risk addressed.

Line 205, What was used to record data and where was data stored.

Under 226, I suggest authors remove the social support needs from the general social network. That is, the first section of social support needs can be deleted, and the results presents the social support needs from participants male partners and the period they needed the support in order to align with the strong assertions made in the background.

558 most participants are undergraduates and what implications does this have to the present study. The socio-economic background of participants is something that could have been explored from the participants. What recommendations can be made to deal with students emotional risk that the lack of male partner support impose on unmarried pregnant students. Additionally, what strategies can be recommended to cultivate a culture of male social support towards their female pregnant partners.

6. PLOS authors have the option to publish the peer review history of their article (what does this mean?). If published, this will include your full peer review and any attached files.

Reviewer #1: **Yes: **Sogo F Matlala

Reviewer #2: No

---

## [Author Response · Author response to Decision Letter 0]

4 Feb 2023

Reviewer #1: Research ethics: It is indeed important to observe research ethics but this does not justify your statement that data cannot be shared publicly because of privacy and confidentiality issues. You can anonymize the transcripts and make them publicly available. Already you have used pseudonyms to protect the identity of participants (Line 210 and Table 1).

We have provided the minimal underlying data as a supporting (supplementary) information file. 

You have correctly indicated (Line 166) that 'issues relating to pregnancy among young people are sensitive' but you have revealed the identity of the institution where data was collected (Lines 160, 161 & 163). I suggest that the name of the university be concealed.

Thank you for this suggestion. We have not revealed the name of the University where the study was conducted in the manuscript. 

Participants: The study is among unmarried pregnant university students as you clearly indicated in the title (Line 1) and some other places in the manuscript. I find it confusing when you refer to participants as 'pregnant women'; 'pregnant young women'; 'unmarried pregnant students'; young unmarried women' in Lines 46, 147, 321, 555 and at some other places. I suggest that a concept to refer to participants be identified and be used consistently. Looking at the age of the participants (Table 2), is it appropriate to refer to all of them as young women? 

Thank you for this comment. We have chosen to refer to participants as “unmarried pregnant students” throughout the manuscript. 

I suggest that Table 1 be moved to Results as it reflects what the study found.

We have moved Table 1 to the Findings section as suggested. 

It is a good practice to describe the qualifications and experiences of researchers in a qualitative research. The description (Lines 174 - 179) of the first author (TMP) is correct, and I suggest that the other authors be described as well.

We have added a brief description of the qualifications and experiences of all the researchers who collaborated in the development of the manuscript. For example: “OA is a Public Health Promotion and Research Methods specialist” and “PN; a post-doctoral fellow and researcher”. 

Language: Replace Master's (Lines 175 - 177) with Masters. 

We have made the correction and replaced Master's with Masters.

School (Lines 242, 482) should be replaced with university.

Thank you for this correction. We have used the term “university” in line 245 and “tuition” in line 485.

The manner of writing ANC (Lines 527, 536 and 551) should rectified. Write in full with abbreviation in brackets on first time use, then abbreviation thereafter. 

We have rectified this error and also rectified in the different sections of the manuscript. 

The concept 'clinics' (Lines 536 - 538) needs to be clarified as specialists run clinics in both private and government/public health services. 

Thank you for this suggestion. We have clarified the concept “clinics” and explained that, “Some participants explained that they were not attending ANC in public clinics but instead were consulting specialists like gynaecologists and obstetricians in private hospitals to monitor their pregnancy. This was mostly because they felt that they needed to be checked by specialists who were not readily available in public clinics”. 

Lines 551 - 557 regarding ANC need to be rephrased as ANC is available through private and public health facilities. 

We have revised this section and removed some material as it was not relevant. This section now reads: “Overall, it was established that most unmarried pregnant students seek ANC in public clinics and hospitals alone without the support of their male partners. The participants generally considered ANC to be unfriendly. In addition, partner attendance was not specifically encouraged or recommended by healthcare workers”. 

Furthermore, pregnant women can deliver in a public hospital even when they did not attend ANC at a public/government facility or even when they did not attend ANC at all. Therefore, Lines 668 - 669 should be revised. 

Thank you for this correction. We have amended the section and explained that, “The South Africa Department of Health guidelines recommend that pregnant women seek ANC as soon as they suspect a pregnancy and encourages all pregnant women to attend at least eight ANC visits”. 

Line 553 - 554 contradicts itself regarding affordability of private healthcare (none & except), see also Lines 670 - 671 regarding ability to pay for private healthcare. 

We have removed this contradicting sentence.

Pregnancy and conception (Line 639) should be conception and pregnancy.

Thank you for this correction, we have amended to highlight the suggested change. 

Description of study design (Lines 150 - 158) should be rephrased to clarify if qualitative research is a design or a method. Based on what you wrote in Line 157, the appropriate design appears to be a narrative design or even a phenomenological design might be suitable.

We have rephrased in the manuscript and included “descriptive” qualitative research design. There are a number of researchers who believe and support the fact that ‘descriptive qualitative research design’ is a viable and acceptable research design (Doyle et al., 2020; Lambert & Lambert, 2012; Ulin et al., 2005). This design is an acceptable research design for robust scholarly research and has received varying degrees of acceptance within the academic community. In this study we use the descriptive qualitative research as a design in its own right. The most frequently proposed rationale for the use of a descriptive design is to provide straightforward descriptions of experiences and perceptions (Sandelowski, 2000), particularly in areas where little is known about the topic under study.

Doyle, L., McCabe, C., Keogh, B., Brady, A., & McCann, M. (2020). An overview of the qualitative descriptive design within nursing research. Journal of Research in Nursing, 25(5), 443-455.

Lambert, V. A., & Lambert, C. E. (2012). Qualitative descriptive research: An acceptable design. Pacific Rim International Journal of Nursing Research, 16(4), 255-256.

Ulin, PR, Robinson, ET, & Tolley, EE (2005). Qualitative methods in public health: A field guide for applied research. Jossey-Bass.

Sandelowski M. Whatever happened to qualitative description?. Research in nursing & health. 2000;23(4):334-40.

Indicate under Data collection if field notes were collected or not as they are important to add to the context of the interviews.

We have explained that, “The interviewer (TMP) made field notes during the interview and then summarized these following the interview. The field notes were thereafter used to develop analytical memos.”.

Data analysis: Indicate (Line 209) that data was transcribed verbatim. This is only indicated in the Abstract (Line 39), but it is important to indicate this in the main document.

We have highlighted in the manuscript that data was transcribed verbatim. The sentence now reads, “The audio-recorded data was transcribed verbatim by the first author (TMP)”.

Results: Indicate the number of sub-themes in Line 229. 

We have indicated the number of sub-themes. “The findings are organized into four main themes and ten sub-themes identified from the analysis of the data”. 

Line 239 should say sub-themes, not themes. Clarify also 'four main subthemes arose 

We have amended to read, “The following three sub-themes emerged from the interviews.”. 

(Line 334). Is the word 'main' necessary? 

We have removed the word. 

The following concepts [many of the participants; several participants; most participants; a few of the participants] are not appropriate to report qualitative findings as they suggest numbers; appropriate concepts are some participates, other participants and others. Writing just 'participants' may also imply that it is all participants who shared that experience. I found both appropriate and inappropriate concepts in this manuscript and suggest that the inappropriate concepts be replaced.

Thank you for this comment. We have used the suggested qualifiers in some places where possible, but in some instances, we felt that some of the qualifiers we had used were the most appropriate way to convey the general experiences of the participants in relation to a particular phenomenon.

Line 311 should acknowledge that some participants had previous pregnancies and therefore knew what to expect. 

We have amended this section to read, “One of the participants who had a previous pregnancy explained the difficulties of having a first pregnancy without adequate knowledge of the accompanying physical changes”.

Lines 319 - 321 and 330 - 331 appears to be misplaced; they would suit the Discussion section.

We have removed the sentence in line 319-321. In line 330-331 we have amended the sentence to show our interpretation of the participant’s experience. The sentence now reads, “The participant’s narrative highlights the challenges of poor nutritional intake during pregnancy”.

Context: It is important to mention if these pregnancies were planned or not, if participants disclosed their pregnancies to whoever they expected to give social support and whether they are still in a continuing and healthy relationships with the partner who impregnated them. Where (on-campus or off-campus) and with whom were participants staying? Lines 599, 606-607 are not clear regarding where participants were staying.

In the design and data collection of the study we did not specifically seek to explore information on the nature of the pregnancies (planned or unplanned); whether they were staying on-campus or off-campus; and with whom they were staying. However, some of this information emerged during the course of our interviews and hence it is reflected in some of the narratives. In instances where this information emerges we attempt to provide some context. Pregnancies were disclosed in all cases to significant others who were expected to offer social support and we have highlighted this in the manuscript.

Discussion: Lines 590 and 648-649 give an impression that types of social support were rated in this study. Rating is a quantitative concept. 

The study sought to explore the social support needs of unmarried pregnant students in tertiary institutions during pregnancy and the puerperal period, and provide insight into the types of social support that were most important to them (primary source of support). However, we have revised the language such that it does not reflect quantitative concepts: 

“Although a previous study reported on the importance of emotional support during pregnancy, it was not rated regarded as the most important type of support needed”.

“Partners’ instrumental and financial supports were also considered to be important sources of support”.

Lines 653-656 should say some African cultures because in some communities in South Africa, paying damages means admission that one has impregnated a woman by paying a fine and not a commitment to support the child financially (paying damages is a once-off payment). 

Thank you for this correction. We have specified that: “In some African cultures, payment of damages is perceived as a form of acceptance of paternity. This process was perceived by the participants to give a form of dignity to her family and some assurance that the father was prepared to take care of the child”. 

Recommendation: It is not clear on which findings are the recommendations on behaviour change programme (Line 688) and rights-based approach (Line 639) based. 

In the manuscript we explain that the participants mentioned that some male partners were not usually willing to help them with tasks. Experiences of social stigma were also reported at public health facilities. Hence, to overcome some of these challenges’ interventions informed by behaviour change and rights-based approach are essential. We explain in the manuscript that:

“Absence or lack of emotional social support from significant others, particularly partners may exacerbate the stressors of pregnancy among students. Tertiary institutions should partner with established social initiatives offering services for men in the country to engage and mentor young men one-on-one and in groups about the importance of the father’s role in the child’s development and how to best support their partners to improve pregnancy outcomes, irrespective of relationship status. Culturally sensitive health promotion programmes targeted at ‘father inclusive’ practices and behaviour change communication programmes to stimulate social support may have positive changes in terms of the supportive role of partners”. 

“Male involvement programmes can be integrated in ANC whereby services are restructured to accommodate male partners to ensure a sense of involvement and responsibility. However, for these programmes to work, the ANC environment should be user friendly for young women and their partners. A rights-based approach must be consistently adopted when delivering maternal healthcare. Training and on-going education should be prioritised for healthcare workers regarding the provision of positive, respectful and supportive non-judgmental antenatal and postnatal care to their clients irrespective of their age or marital status”. 

Who are the 'antenatal health personnel' in Line 695?

We have amended the sentence and explained that: “Training and on-going education should be prioritised for healthcare workers regarding the provision of positive, respectful and supportive non-judgmental antenatal and postnatal care to their clients irrespective of their age or marital status.”

Reviewer #2: There is need to rephrase the topic to experiences of unmarried pregnant students on male partner social support to speak to the background of the study.

Thank you for this comment. We have revised the background to highlight that the main issue of concern is social support needs for unmarried pregnant students and not only male partner social support. While we acknowledge that male social support is one of the main themes that emerged in the findings, the focus of the study is social support in general. However, we have rephrased the topic to, “Perspectives about social support among unmarried pregnant university students in South Africa”. 

On abstract the section of methods line 35, the candidates say perceptions there must be consistency in usage of words, what is explored are experiences.

We have rephrased to clearly highlight that the study is exploring perspectives about social support among unmarried pregnant students in a university setting and not experiences. 

40-41 the results should be reported in line with the study aim which is pregnant women support needs and when it was needed the most.

We have revised this section to highlight the aim of the study which is, “to explore the perspectives of social support among unmarried students in a university setting during pregnancy and the puerperium; and the period when most support is needed”. The results are reported in line with this study aim.

Under line 150, it should be clear which research design under qualitative research approach was employed that informed the thematic content analysis.

We have rephrased in the manuscript and included “descriptive” qualitative research design. There are a number of researchers who believe and support the fact that ‘descriptive qualitative research design’ is a viable and acceptable research design (Doyle et al., 2020; Lambert & Lambert, 2012; Ulin et al., 2005). This design is an acceptable research design for robust scholarly research and has received varying degrees of acceptance within the academic community. In this study we use the descriptive qualitative research as a design in its own right. The most frequently proposed rationale for the use of a descriptive design is to provide straightforward descriptions of experiences and perceptions (Sandelowski, 2000), particularly in areas where little is known about the topic under study. In this type of design, data analysis “does not use a pre-existing set of rules that have been generated from the philosophical or epistemological stance of the discipline that created the specific qualitative research approach. Rather, qualitative descriptive research is purely data-derived in that codes are generated from the data in the course of the study” (Lambert & Lambert, 2012, p.256). Thus, we have further explained in the manuscript that. “As qualitative descriptive research is purely data-derived, we specifically employed an inductive approach to thematic analysis”. 

Under 171 how many students were snowballed were in puerperal period.

Among the first group of students who were approached, all of them were pregnant. None of them were in the puerperal period. However, through snowball sampling we were able to identify these students who had recently given birth. Essentially, all of the participants in the puerperal period were recruited through snowball. In the manuscript we explain that: “The first author (TMP) initially approached three pregnant young students who stayed in the same residence as her. She (TMP) discussed about the study with those young pregnant students and requested them to identify and recommend other students who were pregnant or had recently given birth”.

Line 145 should specify the experiences from unmarried pregnant students are about "male social support".

We have explained earlier on that the study was not specifically focusing on “male partner support” but rather support needs of unmarried pregnant students. We have explained in the manuscript that, “Therefore, the aim of this study is to explore the perspectives about social support among unmarried pregnant students in a university setting”. 

119-144 provides an impression that the social support explored is only from male partners. Line 147 emphasis the said point which contradicts the study topic.

We have revised this section and deleted some material which suggested that the social support explored is only from male partners. 

On line 197 the interview indicates that the schedule elicited the support needs of pregnant students and the period when most support is needed. The results should be aligned to what the interview guide elicited

We have revised this section to include all the topics covered in the interview schedule. This section now reads, “The questions contained in the interview schedule included the socio-demographic variables of participants and their partners, the support needs of pregnant students, the period when most support is needed and types of male partner support needed”. 

on line 149, the section on study objectives should be included.

We have included the objectives of the study:

1) To explore the social support needs of unmarried students in tertiary institutions during pregnancy and the puerperium.

2) To explore the period when most support is needed by unmarried students in tertiary institutions during pregnancy and the puerperium.

line 171-173, how many peripheral students were identified, only pregnant student are mentioned.

Among the first group of students who were approached none of them were in puerperal period and we only managed to gather these participants in the puerperal period through referral. We have explained that: “The first author (TMP) initially approached three pregnant young students who stayed in the same residence as her. She (TMP) discussed about the study with those young pregnant students and requested them to identify and recommend other students who were pregnant or had recently given birth”. 

Under 189, what ethical codes were considered in this study and the study is sensitive in nature, how was the emotional risk addressed.

We explained that: “Data collection was carried out in accordance with relevant guidelines and the participants could withdraw from the study at any point. Although the study was of a sensitive nature, we did not anticipate any negative effects on the participants. However, we had arranged counselling services for the participants if the need arises but none of the participants requested for these services”.

Line 205, What was used to record data and where was data stored.

We have explained that: “We used a digital audio voice recorder and the recordings were safely stored in a password protected computer”.

Under 226, I suggest authors remove the social support needs from the general social network. That is, the first section of social support needs can be deleted, and the results presents the social support needs from participants male partners and the period they needed the support in order to align with the strong assertions made in the background.

Thank you for this suggestion. We have revised the background to highlight that the study will explore social support needs of unmarried pregnant students and not primarily focus on the social support needs from participants male partners. We believe that removing the social support needs from the general social network will not do justice to the study as these are some of the main issues that were expressed by the participants. 

558 most participants are undergraduates and what implications does this have to the present study. The socio-economic background of participants is something that could have been explored from the participants

We did not specifically seek to explore the influence of the socio-economic background of participants nor did we aim to understand the implications of the level of study on the experiences of the participants. We acknowledge that these aspects could have provided a different perspective and insight to the study’s findings. However, there are sections where we elaborate on the influence of the participants’ socio-economic background. For example, “The baby needs so many things so as to grow in a good state; nappies, warm clothes, milk, formula [baby food] since am not going to breastfeed so financially it will be much required from him. [She gives an example]-cause my cousin gave birth in April and the father of the baby hasn’t given her a single cent since she gave birth, since she fell pregnant, she hasn’t received any money from the baby’s daddy and telling me things are difficult, so financially it is so important”(pg. 20, line 438-443); “Lastly, financial support was needed to buy food, to cover transport costs and medical expenses and to prepare for the arrival of the unborn baby” (pg. 28, ;line 611-613). 

What recommendations can be made to deal with students’ emotional risk that the lack of male partner support impose on unmarried pregnant students. Additionally, what strategies can be recommended to cultivate a culture of male social support towards their female pregnant partners.

Thank you for highlighting these issues. As part of the recommendations we have added that, “In instances where emotional support from their male partners is not feasible, unmarried pregnant students need to identify alternate sources of support such as counselling support, friends, church and social media”. 

“Tertiary institutions should partner with established social initiatives offering services for men in the country to engage and mentor young men one-on-one and in groups about the importance of the father’s role in the child’s development and how to best support their partners to improve pregnancy outcomes, irrespective of relationship status. Culturally sensitive health promotion programmes targeted at ‘father inclusive’ practices and behaviour change communication programmes to stimulate social support may have positive changes in terms of the supportive role of partners. Implementors of such programmes should consider adopting innovative outreach strategies that appeal to young men’s interests and hang-out spots, such as bars and sports arenas”.

---

## [Decision Letter · Decision Letter 1]

6 Mar 2023

PONE-D-22-16990R1Perspectives about social support among unmarried pregnant university students in South AfricaPLOS ONE

Dear Dr. Akintola,

Thank you for submitting your manuscript to PLOS ONE. After careful consideration, we feel that it has merit but does not fully meet PLOS ONE’s publication criteria as it currently stands. Therefore, we invite you to submit a revised version of the manuscript that addresses the points raised during the review process. Many thanks to the authors for this revised manuscript. This is a really impressive piece of work for a master's study--really well done! I think it is reading a lot more clearly now. I have made a number of comments in the text directly (please see your manuscript attached). Many are typographical edits, but some pertain to the contents.

Additionally, there are a some overarching areas that I would like to see revised:

- The introductory section seems to assume that all university students who get pregnant are not married, likely in an unstable relationship with their partner, and that the pregnancy is unintended. This is not always going to be the case, so you may want to caveat this as “unmarried women with unintended pregnancies”, for example. At the very least, there needs to be a bit more nuance here, acknowledging that women get pregnant under different circumstances, including for women in university.

- It would also be important to briefly mention the legality of abortion—you note that many students choose to terminate their pregnancies, but this is of course only likely/safe where abortion is, so if you could add a sentence about abortion within the South African context, that would be helpful. Although it is legal, I imagine it still carries certain stigma (and potentially different access barriers, making it an unavailable service for all women), so there may be women who are likely to continue with their pregnancy, even though they may prefer to terminate. In which case implications for social support are likely different when the pregnancy, though unintended, is not wanted, versus where it is wanted. As you note in your results, social supports like partner support mediate this (and the decision to terminate, if that is an option).

- I would ask that you broadly streamline the introduction, as there is quite a bit of repetition (social stigma, relationship with partner, etc. are raised multiple times).

- In your methods, please given an indication of the types of questions that were asked.

- Also, it would be useful to add if you inevitably sampled until theoretical saturation had been reached, which would suggest an added level of rigour.

- Would it be possible to add to Table 2 if students were undergraduate or graduate students? As it’s clear you did recruit graduate students from the initial three, the breakdown by years seems to suggest only undergraduate students, which is confusing. It’s quite different to be a second year undergraduate versus a second year PhD student, for example. I think this would be an important detail that may affect their experience (especially given the very different demands at the undergraduate and postgraduate levels).

- I am not clear on how many themes you have! You say three, but then you have a theme about timing, and then about male support (which has its own sub-themes, I’m assuming the health system one is a sub-theme for this?)—so I’d say five? Please clarify.

- Please ensure the information about the participant giving each quotation is consistent (e.g. sometimes it’s age, sometimes it’s pregnancy/postpartum status, sometimes it’s educational level).

- Also ensure that all quotations in your results have a period at the end—I’d recommend inside of the quotation and before the information about the participant, as that is more grammatically correct.

- Where something was not consistently expressed and may be a divergent viewpoint, please make that clear. For example, you quoted one of your participants who was going to have a small baby, which you write as though nutritional challenges were consistent across your participants, but her experience is quite uncommon (for a baby to be small for gestational age to that extent, and especially due to nutrition—she would have had to be quite malnourished for this to be the case as the fetus will take what it needs and generally do well, as long as the maternal nutrition status was okay prior to pregnancy—if it was really poor, she’d likely not be menstruating and ovulating at all).

- For the timing query: I’d call these sub-themes, but I’d be clear about indicating an overarching theme, which, based on your results, seems to be something like, “timing of concentrated support in pregnancy varies depending on the participants’ circumstances”.

- In your discussion, it’d be very interesting to see a reflection on why you feel appraisal support didn’t emerge.

- You have some nice recommendations—it would also be useful to see some suggested areas for further research.

- It would also be important to see a brief reflection on some of the strengths and limitations of your study, specifically your methodological approach, within the discussion. Please submit your revised manuscript by Apr 20 2023 11:59PM. If you will need more time than this to complete your revisions, please reply to this message or contact the journal office at plosone@plos.org. Please include the following items when submitting your revised manuscript:A rebuttal letter that responds to each point raised by the academic editor and reviewer(s). You should upload this letter as a separate file labeled 'Response to Reviewers'.A marked-up copy of your manuscript that highlights changes made to the original version. You should upload this as a separate file labeled 'Revised Manuscript with Track Changes'.An unmarked version of your revised paper without tracked changes. You should upload this as a separate file labeled 'Manuscript'.If applicable, we recommend that you deposit your laboratory protocols in protocols.io to enhance the reproducibility of your results. Protocols.io assigns your protocol its own identifier (DOI) so that it can be cited independently in the future. For instructions see: https://journals.plos.org/plosone/s/submission-guidelines#loc-laboratory-protocols. Additionally, PLOS ONE offers an option for publishing peer-reviewed Lab Protocol articles, which describe protocols hosted on protocols.io. Read more information on sharing protocols at https://plos.org/protocols?utm_medium=editorial-email&utm_source=authorletters&utm_campaign=protocols.

We look forward to receiving your revised manuscript.

Kind regards,

Tara Tancred, PhD

Academic Editor

PLOS ONE

Journal Requirements:

Reviewers' comments:

Reviewer's Responses to Questions

**Comments to the Author**

1. If the authors have adequately addressed your comments raised in a previous round of review and you feel that this manuscript is now acceptable for publication, you may indicate that here to bypass the “Comments to the Author” section, enter your conflict of interest statement in the “Confidential to Editor” section, and submit your "Accept" recommendation.

Reviewer #1: All comments have been addressed

2. Is the manuscript technically sound, and do the data support the conclusions?

Reviewer #1: Yes

3. Has the statistical analysis been performed appropriately and rigorously? 

Reviewer #1: N/A

4. Have the authors made all data underlying the findings in their manuscript fully available?

Reviewer #1: Yes

5. Is the manuscript presented in an intelligible fashion and written in standard English?

Reviewer #1: Yes

6. Review Comments to the Author

Reviewer #1: The authors responded appropriately to reviewers' comments and the manuscript adds value to qualitative research

7. PLOS authors have the option to publish the peer review history of their article (what does this mean?). If published, this will include your full peer review and any attached files.

Reviewer #1: **Yes: **Sogo France Matlala

---

## [Author Response · Author response to Decision Letter 1]

30 Mar 2023

#1: The introductory section seems to assume that all university students who get pregnant are not married, likely in an unstable relationship with their partner, and that the pregnancy is unintended. This is not always going to be the case, so you may want to caveat this as “unmarried women with unintended pregnancies”, for example. At the very least, there needs to be a bit more nuance here, acknowledging that women get pregnant under different circumstances, including for women in university.

Thank you for this comment. We have revised part of the introduction to indicate this caveat that the study focuses on unmarried women with unintended pregnancies. We have included literature that points to the high probability of unmarried university students having unintended pregnancies. We have explained in the manuscript that: “Although social support is of great value for all mothers during pregnancy, certain groups of women are especially vulnerable during pregnancy, and these include adolescents, unmarried women, students and women of low socio-economic status [7]. Because of the risky sexual behavior among students, young women in institutions of higher learning have a high risk of unintended pregnancy [8], particularly among unmarried students [9-11]”.

#2: It would also be important to briefly mention the legality of abortion—you note that many students choose to terminate their pregnancies, but this is of course only likely/safe where abortion is, so if you could add a sentence about abortion within the South African context, that would be helpful. Although it is legal, I imagine it still carries certain stigma (and potentially different access barriers, making it an unavailable service for all women), so there may be women who are likely to continue with their pregnancy, even though they may prefer to terminate. In which case implications for social support are likely different when the pregnancy, though unintended, is not wanted, versus where it is wanted. As you note in your results, social supports like partner support mediate this (and the decision to terminate, if that is an option).

We do agree that abortion may be associated with some stigma. We briefly describe the legality of abortion in the South African context. We explain that, “While some students may choose to terminate their unplanned pregnancies as a legal right stipulated under the Choice in Termination of Pregnancy Act in South Africa [12], abortion is often publicly condemned and associated with negative and judgmental attitudes from health care providers and the community [13, 14]. Students that choose to keep their unplanned pregnancies remain in need of a supportive environment that favours their physical and psychological wellbeing”. 

#3: I would ask that you broadly streamline the introduction, as there is quite a bit of repetition (social stigma, relationship with partner, etc. are raised multiple times).

We have revised the introduction and removed some material which was repetitive. 

#4: In your methods, please given an indication of the types of questions that were asked.

We had broadly included the themes from where the interview questions were drawn. Additionally, we included a few examples of the questions that comprised the interview schedule. In the manuscript we have added that: Some of the interview questions posed to participants include: “Describe some of the challenges you face in your pregnancy? As a pregnant student?”; “At what stage of pregnancy do you feel you need the most support from significant others? Why and from who?”; and “From the support needs you have already mentioned, which ones do you feel are the most important to you? Why?”.

#5: Also, it would be useful to add if you inevitably sampled until theoretical saturation had been reached, which would suggest an added level of rigour.

In the analysis section of the manuscript we have explained that, “Although 16 interviews already indicated a point of saturation in which no new codes were identified, we decided to continue analysis of the remaining interviews to ensure the confidence and verification of data saturation”. 

#6: Would it be possible to add to Table 2 if students were undergraduate or graduate students? As it’s clear you did recruit graduate students from the initial three, the breakdown by years seems to suggest only undergraduate students, which is confusing. It’s quite different to be a second year undergraduate versus a second year PhD student, for example. I think this would be an important detail that may affect their experience (especially given the very different demands at the undergraduate and postgraduate levels).

Thank you for this suggestion. We have made the level of study of participants clearer by adding to the table that all 1st, 2nd and 3rd year participants were undergraduates. Only those who were doing an honours degree were post-graduates (or graduate) students. 

#7: I am not clear on how many themes you have! You say three, but then you have a theme about timing, and then about male support (which has its own sub-themes, I’m assuming the health system one is a sub-theme for this?)—so I’d say five? Please clarify.

We explain in the findings that, “The findings are organized into four main themes and ten sub-themes identified from the analysis of the data. The four main themes are: Support needs of unmarried pregnant students; period when the most support is needed; male partner support required; and health systems role on social support”.

For clarity, we have listed the themes and sub-themes below:

Support needs of unmarried pregnant students: emotional support; instrumental support; informational support

Period when the most support is needed: at the beginning of pregnancy; throughout pregnancy; towards the end of pregnancy; support post delivery

Male partner support required: partner’s emotional support; partner’s instrumental support; partner’s financial support

Health systems role on social support

#8: Please ensure the information about the participant giving each quotation is consistent (e.g. sometimes it’s age, sometimes it’s pregnancy/postpartum status, sometimes it’s educational level).

We have provided the pregnancy and postpartum status of all participants and for those who have a previous pregnancy we include this detail for context. We also include the educational level of participants broadly as undergraduate or post-graduate. 

#9: Also ensure that all quotations in your results have a period at the end—I’d recommend inside of the quotation and before the information about the participant, as that is more grammatically correct.

We have gone through the results and ensured that all quotations have a period at the end. 

#10: Where something was not consistently expressed and may be a divergent viewpoint, please make that clear. For example, you quoted one of your participants who was going to have a small baby, which you write as though nutritional challenges were consistent across your participants, but her experience is quite uncommon (for a baby to be small for gestational age to that extent, and especially due to nutrition—she would have had to be quite malnourished for this to be the case as the fetus will take what it needs and generally do well, as long as the maternal nutrition status was okay prior to pregnancy—if it was really poor, she’d likely not be menstruating and ovulating at all).

Thank you for this comment. We have made it clear that the issue of nutritional challenges was unique. In the manuscript we have revised to read, “In a unique case, one of the participants was at risk of adverse pregnancy outcomes as she lacked adequate information on proper nutrition practices as illustrated below”. 

#11: For the timing query: I’d call these sub-themes, but I’d be clear about indicating an overarching theme, which, based on your results, seems to be something like, “timing of concentrated support in pregnancy varies depending on the participants’ circumstances”.

The editor’s earlier comment was about the need to make the themes and sub-themes clear, which we have done. We do believe that the present themes and sub-themes we use correctly capture the essence of the study findings. With regards to this specific theme we had named it “Period when the most support is needed”, with the following sub-themes: at the beginning of pregnancy; throughout pregnancy; towards the end of pregnancy; and support post-delivery.

#12: In your discussion, it’d be very interesting to see a reflection on why you feel appraisal support didn’t emerge.

Thank you for this suggestion. We have added in the discussion that, “However, appraisal support did not emerge as an important source of social support in this study and this could be because most of the participants were still pregnant and felt they did not need evaluative feedback. Yet, this form of support could have been of great value to them after delivery as they engage in motherhood duties such as feeding and bathing the infant”. 

#13: You have some nice recommendations—it would also be useful to see some suggested areas for further research.

Thank you for this suggestion. We added suggested areas for further research, “Future studies should include different racial groups to capture the experiences of pregnant students of all races and also include married and unmarried students to compare and contrast their needs.”

#14: It would also be important to see a brief reflection on some of the strengths and limitations of your study, specifically your methodological approach, within the discussion.

Thank you for this suggestion. We added the strengths and limitations of the study:

“To our knowledge, this is the first study to present a rich description of the perspectives about social support needs and the period when most support is needed by unmarried pregnant university students in South Africa. We also sampled until data saturation to ensure that a sufficient quantity and quality data was collected. However, the snowball method of sampling that was used meant that the initial participants, who were indigenous African students, recommended other pregnant students in their social network, who were also indigenous African students. As a result, potential participants from other racial groups did not participate in this study. Therefore, the findings could be biased and be a representation of experiences and perceptions of pregnant indigenous African students. Also, the findings of this study apply only to pregnant students who are unmarried and does not apply to those pregnant students who are in a marriage or engaged to be married. Future studies should include different racial groups to capture the experiences of pregnant students of all races and also include married and unmarried students to compare and contrast their support needs”.

---

## [Editor Report · Decision Letter 2]

12 Apr 2023

Perspectives about social support among unmarried pregnant university students in South Africa

PONE-D-22-16990R2

Dear Dr. Akintola,

We’re pleased to inform you that your manuscript has been judged scientifically suitable for publication and will be formally accepted for publication once it meets all outstanding technical requirements.

Kind regards,

Tara Tancred, PhD

Academic Editor

PLOS ONE

Additional Editor Comments (optional):

Happy to see this manuscript accepted. Thank you to the authors for their hard work in getting it to this stage. I think this will make a useful contribution to literature.
---

## [Editor Report · Acceptance letter]

14 Apr 2023

PONE-D-22-16990R2 

Perspectives about social support among unmarried pregnant university students in South Africa 

Dear Dr. Akintola:

I'm pleased to inform you that your manuscript has been deemed suitable for publication in PLOS ONE. Congratulations! Your manuscript is now with our production department. 

Kind regards, 

on behalf of

Dr. Tara Tancred 

Academic Editor

PLOS ONE